

# The effects of allogenic stem cells in a murine model of hind limb diabetic ischemic tissue

Jesús Álvarez García[1], Soledad García Gómez-Heras[2], Luis Riera del Moral[3], Carlota Largo[3], Damián García-Olmo[4,5] and Mariano García-Arranz[4,6]

[1] Vascular Surgery, Hospital Quirón San Camilo, Madrid, Spain
[2] Department of Human Hystology, Health Science Faculty, Rey Juan Carlos University, Alcorcón, Madrid, Spain
[3] Experimental Surgery Department, Hospital Universitario La Paz, Madrid, Spain
[4] Department of Surgery, Universidad Autónoma de Madrid, Madrid, Spain
[5] Department of Surgery, Hospital Universitario Fundación Jimenéz Díaz, Madrid, Spain
[6] New Therapies Lab, Instituto de Investigación Sanitaria Fundación Jiménez Díaz, Madrid, Spain

Corresponding author
Soledad García Gómez-Heras,
soledad.garcia@urjc.es

## ABSTRACT

**Background**. Diabetes is one of the major risk factors for peripheral arterial disease. In patients in whom surgery cannot be performed, cell therapy may be an alternative treatment. Since time is crucial for these patients, we propose the use of allogenic mesenchymal cells.

**Methods**. We obtained mesenchymal cells derived from the fat tissue of a healthy Sprague-Dawley rat. Previous diabetic induction with streptozotocin in 40 male Sprague-Dawley rats, ligation plus left iliac and femoral artery sections were performed as a previously described model of ischemia. After 10 days of follow-up, macroscopic and histo-pathological analysis was performed to evaluate angiogenic and inflammatory parameters in the repair of the injured limb. All samples were evaluated by the same blind researcher. Statistical analysis was performed using the SPSS v.11.5 program ($P < 0.05$).

**Results**. Seventy percent of the rats treated with streptozotocin met the criteria for diabetes. Macroscopically, cell-treated rats presented better general and lower ischemic clinical status, and histologically, a better trend towards angiogenesis, greater infiltration of type 2 macrophages and a shortening of the inflammatory process. However, only the inflammatory variables were statistically significant. No immunological reaction was observed with the use of allogeneic cells.

**Discussion**. The application of allogeneic ASCs in a hind limb ischemic model in diabetic animals shows no rejection reactions and a reduction in inflammatory parameters in favor of better repair of damaged tissue. These results are consistent with other lines of research in allogeneic cell therapy. This approach might be a safe, effective treatment option that makes it feasible to avoid the time involved in the process of isolation, expansion and production of the use of autologous cells.

## INTRODUCTION

Diabetes mellitus is a major risk factor for the development of atherosclerosis (*Hossain, Kawar & Nahas, 2007*), which is the leading cause of death in diabetic patients, the cause of up to 50% of deaths among these patients (*Stamler et al., 1993*).

The involvements of the arteries of the lower extremities, as well as coronary or cerebrovascular disease, are different manifestations of systemic arteriosclerosis. This condition, called peripheral vascular disease, affects 4.3% of the population, and up to 7.5% of diabetics between 60 and 64 years of age (*Norgren et al., 2007*; *Hirsch et al., 2006*).

Current treatment of this condition is revascularization, either through "open" surgery (bypass, plasties and endarterectomies, or "endovascular" procedures: angioplasty and stenting). However, these treatments may not always be possible because of the anatomical characteristics of the lesions and, sometimes, they fail to bring more blood to the ischemic territory. In this situation, as a last option, amputation should be performed. Major amputations are very hard on patients, have high postoperative morbidity and mortality rates, and cause severe deterioration in the quality of life, as well as high healthcare costs (*Bradbury et al., 2010*; *Hiatt, 2001*; *Simons & Ware, 2003*).

In this context, therapeutic angiogenesis targets the proliferation of collateral vessels. Many angiogenic molecules have been described in both animal and human models with ischemic diseases (*Freedman & Isner, 2002*; *Heijnen & Van der Sluijs, 2015*; *Domouzoglou et al., 2015*). The use of different stem cells such as those derived from hematopoietic tissue, fetal membrane and mesenchymal tissue has also been postulated (*Ishikane et al., 2008*; *Shibata et al., 2008*; *Salazar Álvarez et al., 2016*).

Among the mesenchymal cells, stem cells derived from adipose tissue (ASCs) are interesting candidates for this purpose because they can be obtained in a non-aggressive way, have a privileged immunological profile, are multipotent, and have shown safety in both autologous and allogeneic use, in both experimental and clinical trials in Phases I/IIa, for the treatment of ischemic diseases (*Zuk et al., 2002*). ASCs can secrete angiogenic, immunomodulatory, and cellular survival factors that have been shown to be effective in the treatment of atherosclerosis and diabetic complications both in animal and human models (*Iwase et al., 2005*; *Shibata et al., 2008*; *Kim et al., 2006*; *Salazar Álvarez et al., 2016*; *Zuk et al., 2002*). Mesenchymal cells derived from subjects with cardiovascular and diabetic disease are less effective (*Capla et al., 2007*; *Loomans et al., 2004*; *Thum et al., 2007*). Therefore, it seems advisable to use ASCs derived from healthy subjects for the treatment of cardiovascular diseases.

The aim of this study was to analyze the evolution of diabetic rats with hind limb ischemia treated with allogeneic ASCs.

## MATERIALS/METHODS

We used 41 animals, one of which was used to obtain the mesenchymal stem cells from abdominal adipose tissue according to the protocol described by P Zuk and modified by our group (*García-Olmo et al., 2003*). Diabetic induction was performed on 40 male Sprague-Dawley rats from an authorized supplier (Charles River/Janvier), weighing between 152.5
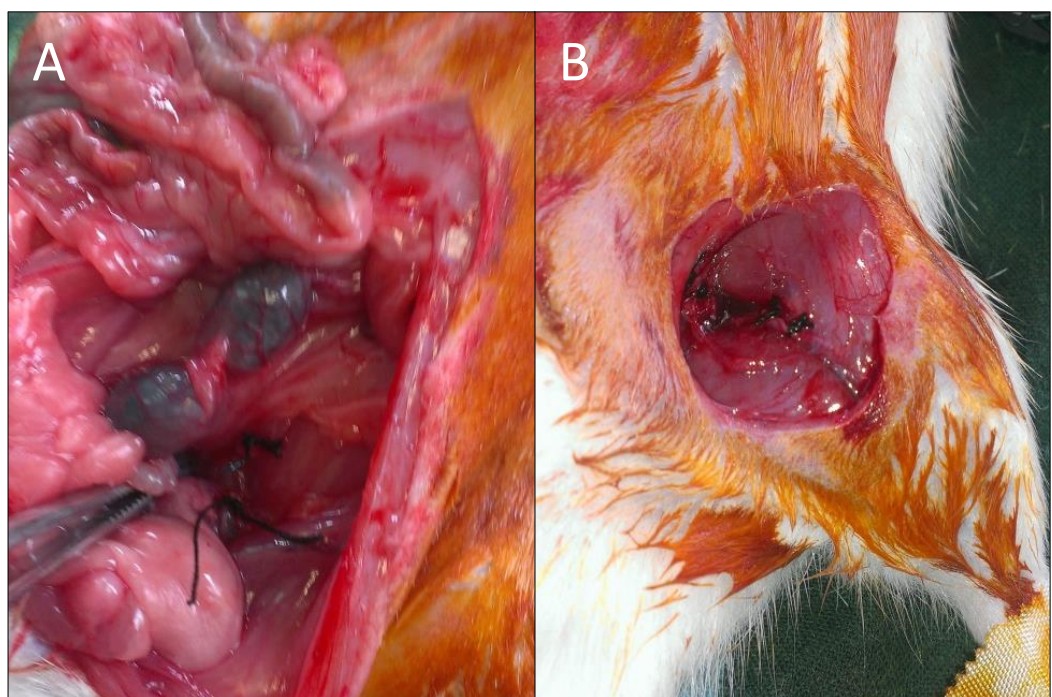

**Figure 1** Images of the surgical procedure.

and 250 g, which were housed at the University Hospital La Paz (280790001941) in groups of 5 specimens with free access to food and drink and under standardized conditions of temperature (24 $\pm$ 1 °C), humidity (55 $\pm$ 5%), and 12-hour light cycles (from 8:00 to 20:00 h). The protocol approved by the Animal Welfare Ethics Committee (No. CEBA/12-14) was followed and set out in the EU Directive on experimental animals (63/2010 EU) and Spanish legislation (RD 53/2013).

Two groups were blinded: *control group* (diabetic animals with hind limb ischemia without ASCs), and *treatment group* (diabetic animals with hind limb ischemia treated with ASCs). Follow-up lasted for 10 days.

Induction with streptozotocin was performed following the protocol of *Wu & Huan (2008)*, and animals were considered diabetic (DM) if they had blood glucose higher than 200 mg/dl fasting. Ten days after induction with streptozotocin, blood extraction was performed to check blood glucose levels and the success of DM generation as a first step.

On animals with glycemia greater than 200 mg/dl, ischemia in the left limb was induced by ligation of the common iliac artery in the aortic bifurcation and the common femoral artery prior to the saphenous-femoral junction with 5/0 silk suture as previously described (*Iwase et al., 2005*; *Paek et al., 2002*). The procedure was done under general anesthesia with 4% isofluranoal for induction and 2% for maintenance (Fig. 1A).

ASCs were obtained from the subcutaneous inguinal fat of a non-diabetic rat. After a sufficient number was obtained, they were cryopreserved in independent aliquots ($1 \times 10^6$ cells per cryovial) for each treatment with 10% dimethyl sulfoxide (DMSO) in Fetal Bovine Serum (FBS) by expansion with Dulbecco's modified Eagle's medium (DMEM) + 10%

FBS + 1% penicillin/streptomycim (10.000 U/mL/10.000 μg/mL). Prior to this, they were thawed for 48 h to activate cells and ensure the exact number of viable cells, eliminating possible cellular variability among treatments. Cell cultures were characterized according to the International Federation for Adipose Therapeutics and Science (IFATS) and the International Society for Cellular Therapy (ISCT) (*Bourin et al., 2013*).

Injection of stem cells into the treatment groups was performed in the same procedure of ischemic surgery, maintaining inhaled anesthesia. $10^6$ cells were implanted in five aliquots: two aliquots for muscle tissue adjacent to the common femoral artery ligature, and three aliquots in the musculature surrounding the path of the ischemic vessel (Fig. 1B).

## FOLLOW-UP

Daily monitoring was carried out, collecting the parameters indicated in Table 1, assessing the physical and mechanical aspects of each animal using a numerical value for each parameter as a function of severity. A cut-off point was established for a change in treatment score of 7 points, and a point of no return, or sacrifice, at 17 points or a greater than 15% weight loss over the previous figure on three consecutive days.

### Histological analysis

For histological studies, limb samples of 5 mm$^3$ were fixed in 10% formaldehyde at room temperature, embedded in paraffin and cut into 5-micron-thick slices in a Micron HM360 microtome. Sections were stained with hematoxylin-eosin to evaluate plasm cells and capillaries, toluidine blue for the identification of the mast cells and immunohistochemistry with antibody anti-CD31 for the arterioles, anti-CD68 for macrophages, anti-CD-206 to detect macrophages M2 and anti-CD-19 for B-lymphocytes detection.

All were studied under a Zeiss Axiophot 2 microscope and photographed with an AxiocamHRc camera.

For immunohistochemical studies, histology sections were deparaffinized and rehydrated before endogenous peroxidase activity was blocked with $H_2O_2$ (0.3%) in methanol. The slides were rinsed with PBS and incubated with primary antibodies in a moist chamber at room temperature. The primary antibodies used were: anti-CD31 polyclonal antibody (orb229364; Biorbyt, Cambridgeshire, UK) at a 1:100 dilution, anti-CD68 (MCA341GA; Bio-Rad, Hercules, CA, USA) at 1:100 dilution, anti-CD206 polyclonal antibody orb180464, Biorbyt) at 1:100 dilution and anti-CD19 (BS-0079R; Bioss, Woburn, MA, USA) at 1:500 dilution. The sections were subsequently incubated with biotinylated anti-rabbit IgG and LBA (DAKO) for 25 min at room temperature, rinsed with PBS and immersed for 25 min. in avidin peroxidase. The immunostaining reaction product was developed using diaminobenzidine. Counterstaining was performed with hematoxylin. The specificity of the immunohistochemical procedure was checked by incubation of sections with non-immune serum instead of primary antibody.

Between fifty and twenty contiguous non-overlapping fields (10×, 20× and 40×) per slide from each group were counted and the results were expressed as cells per field (cells/f), or capillaries per field, or arterioles per field.
**Table 1  Physical parameters to be evaluated during animal monitoring.**

| Weight | | No changes | 0 |
|---|---|---|---|
| | | Loss 5–10% | 1 |
| | | Loss 10–15% | 2 |
| | | Loss 15–20% | 3 |
| | | Loss > 0% | 17 |
| General aspect | Coat | Normal | 0 |
| | | Altered | 1 |
| | | Bristly | 2 |
| | Dehydratation | No | 0 |
| | | Mild | 1 |
| | | Moderate | 2 |
| | | Severe | 17 |
| | Activity | Normal | 0 |
| | | Minor | 1 |
| | | Limited | 2 |
| | | Null | 17 |
| Extremity | Color | Normal | 0 |
| | | Cyanotic | 0 |
| | Limp | No | 0 |
| | | MILD | 1 |
| | | Severe | 2 |
| | Contract | No | 0 |
| | | Yes | 2 |
| | Ulcers | No | 0 |
| | | Yes | 2 |
| | Mutilation | No | 0 |
| | | Yes | 6 |

All were quantified and evaluated by the same researcher with no knowledge of the groupings (blinded).

Quantitative variables between groups were analyzed using the corrected chi-square test. The statistical significance level was set at 0.05. Analyses of data were performed with program SPSS v.11.5 (SPSS Inc. Chicago, IL, USA).

## RESULTS

Of the 40 rats to which diabetes was induced, 70% met established diabetes criteria, reducing our $n$ to 29 diabetic animals (Fig. 2).

The thawed and pre-used cells were characterized according to IFATS recommendations, demonstrating that they were mesenchymal stem cells (data not show).

### Macroscopic results

In the analysis of the ischemic clinical scale, we did not find statistically significant overall differences between the control (1.53) and treated groups (1.69), although our analysis

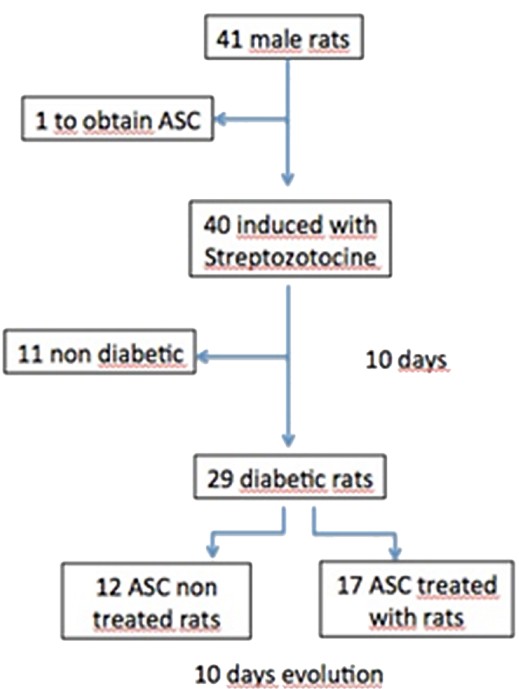

Figure 2 **Flow chart.** ASC, adipose derived stem cells.

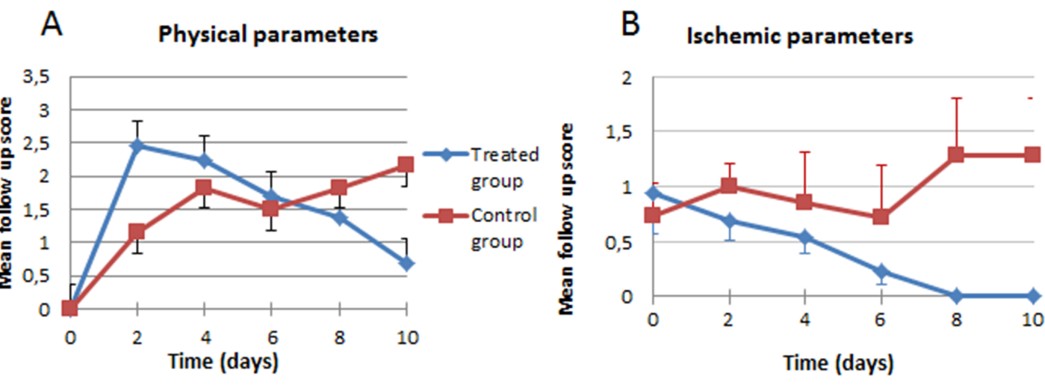

Figure 3 **Clinical evolution.** (A) Physical parameters; (B) ischemic parameters.

of the progression curve objectively showed that, despite starting with a more severe ischemia in the individuals treated with stem cells, they ended up with lower ischemic clinical findings (Fig. 3A). In the general appearance parameters, we found a difference that, though not statistically significant ($p = 0.5453$), was striking because of the tendency to worsen (increase in scales) in the control group, in contrast with the improvement in the group treated with stem cells (Fig. 3B).

However, even though the difference was not significant, at the macroscopic level, the appearance of the paws of the treated animals was better, as can be seen in Fig. 4 (10 days of evolution).

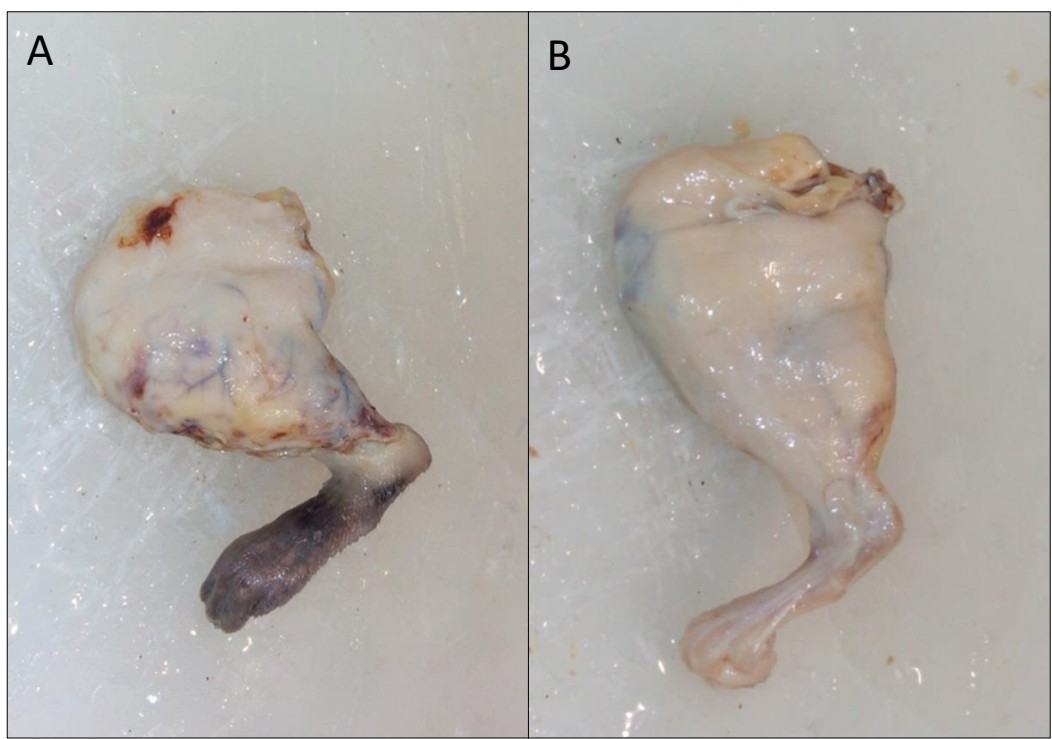

**Figure 4 Macroscopic appearance of the treated limb.** (A) 10 Days after surgery without cellular treatment; (B) 10 Days after surgery and treatment with cells.

## Histopathological results

A typical foreign body reaction (type 1 macrophages) is observed in all cases around the suture.

In the cicatricial zone of the treatment group, a lower inflammatory infiltrate was observed than in the control group ($p < 0.005$), except for plasma cells and mast cells. The neovascularization parameters did not show *statistically significant differences* in arterioles and capillaries, 1.19 vs. 1.39 and 7.71 vs. 12.29 treated group versus control group respectively (Table 2).

In the peri-cicatricial region, the number of macrophage type 2 cells is statistically significant in the treatment group versus the control group ($p < 0.005$) with a high number in treated group. Macrophages are always situated in the peripheral part of the granulation tissue and around the regional vasculo-nervous packages, included in the microglia and the adipose tissue surrounding the region. The neovascularization parameters within the muscle fibers of the peri-cicatricial region show an increase in capillaries in the groups treated with cells (33% more elevate); the remaining vascular parameters do not show significant differences ($p > 0.05$).

In the regional lymph nodes, we also observed differences between the study groups, especially in the number of type 2 macrophages (499 treatment group & 85 control group), with a significant difference ($p < 0.1$), despite a lower number of B- lymphocytes (16.8

Álvarez García et al. (2017), PeerJ, DOI 10.7717/peerj.3664

**Table 2  Evaluation of the histological parameters.**

| | Scar area | | | | | | Attached nodes | | Peri-cicatricial region | |
|---|---|---|---|---|---|---|---|---|---|---|
| | Inflammatory cells | | | | Vascularization | | | | | |
| | No. neutrophils /40× field | No. CD19 + /40× field | No. plasmatic cells /40× field | No. mastocytes /20× field | No. capillaries /40× field | No. arterioles /40× field | No. CD19+ /40× field | No. CD206+ /CD68+ | No. CD206+ /10× field | No. capillaries /50 muscle cells |
| G.1 | 3.18 ± 0.21 | 15.51 ± 0.94 | 17.15 ± 1.70 | 3.1 ± 0.15 | 12.29 ± 0.95 | 1.39 ± 0.11 | 21.3 ± 1.50 | 85 /2,045 | 7.16 ± 0.31 | 24 |
| G.2 | 3.55 ± 0.29 | 14.95 ± 0.45 | 13.12 ± 0.49 | 5.8 ± 0.19 | 7.71 ± 1.29 | 1.19 ± 0.12 | 16.8 ± 1.20 | 499 /2,047 | 10.37 ± 0.43 | 36 |

**Notes.**

G.1, Control group; G.2, Treated group; □, mean ± se = standard error of mean.

lymphocytes/high power field40x in the treatment group& 21.316.8 lymphocytes/high power field40x in the Control group), (Fig. 5).

## DISCUSSION

Peripheral arteriopathy in diabetic patients remains a serious health problem despite the enormous clinical and surgical advances of the last decades. There are patients who cannot benefit from these advances and for them, cellular therapy may be the only alternative to a major amputation. With this study, using allogeneic cells on an experimental model of diabetic rats, we have shown the feasibility of this treatment option and have produced results that support those obtained to date by other authors with autologous mesenchymal cells of any origin.

The results of induction of diabetic animals with streptozotocin in our study are in accordance with previous data published (*Ishikane et al., 2008*; *Iwase et al., 2005*; *Paek et al., 2002*).

The previous models, despite their broad use, have several limitations; the diabetes model attempts to simulate a multivariable pathology, yet it is not applicable to all the different types of diabetes. In addition, it reported a 30% failure, concordant with our results. The produced ischemia is an acute ischemia that does not consider the pathophysiology of atherosclerosis, which is the origin of chronic ischemia.

Most of the studies used to treat cardiovascular disease were conducted with mesenchymal stem cells of different origins (93 trials registered on clinical trial.gov), including 14 with ASCs. In all cases, autologous cells were used.

Since diabetic patients are more likely to develop cardiovascular disease, and we currently know that their ASCs can have impaired functions (*Loomans et al., 2004*; *Capla et al., 2007*), it seems logical to suppose that allogeneic cell therapy (cells from healthy donors) would be more beneficial.

ASCs secrete angiogenic and cell survival factors, and have been shown to be effective in the treatment of coronary disease and the complications of diabetes in animals and human models (*Kim et al., 2006*; *Loomans et al., 2004*; *Williams et al., 2011*; *Capla et al., 2007*). In addition, their immunological characteristics allow for allogeneic transplantation, more practical than autologous transplantation in patients with ischemic pathology with or without associated diabetes, given the time required to obtain enough cells for treatment.

Although in our macroscopical results we have not observed significant overall differences in the clinical scale during follow-up, there is a trend towards improvement in the general situation of the treated animals in aspects including weight and general appearance, as well as the color of the distal end of the affected limb.

The histological results obtained indicate that any alteration between groups is more associated with the secretion of trophic factors by the cells than with the differentiation of these cells, which goes in line with the current knowledge of mesenchymal cell therapies. As for angiogenic parameters, the absence of significant differences in both the cicatricial and peri-cicatricial regions between groups may be due to the small number of animals studied, although a minimal trend is observed, which again agrees with data from other authors (*Katare et al., 2013*; *Shevchenko et al., 2013*).
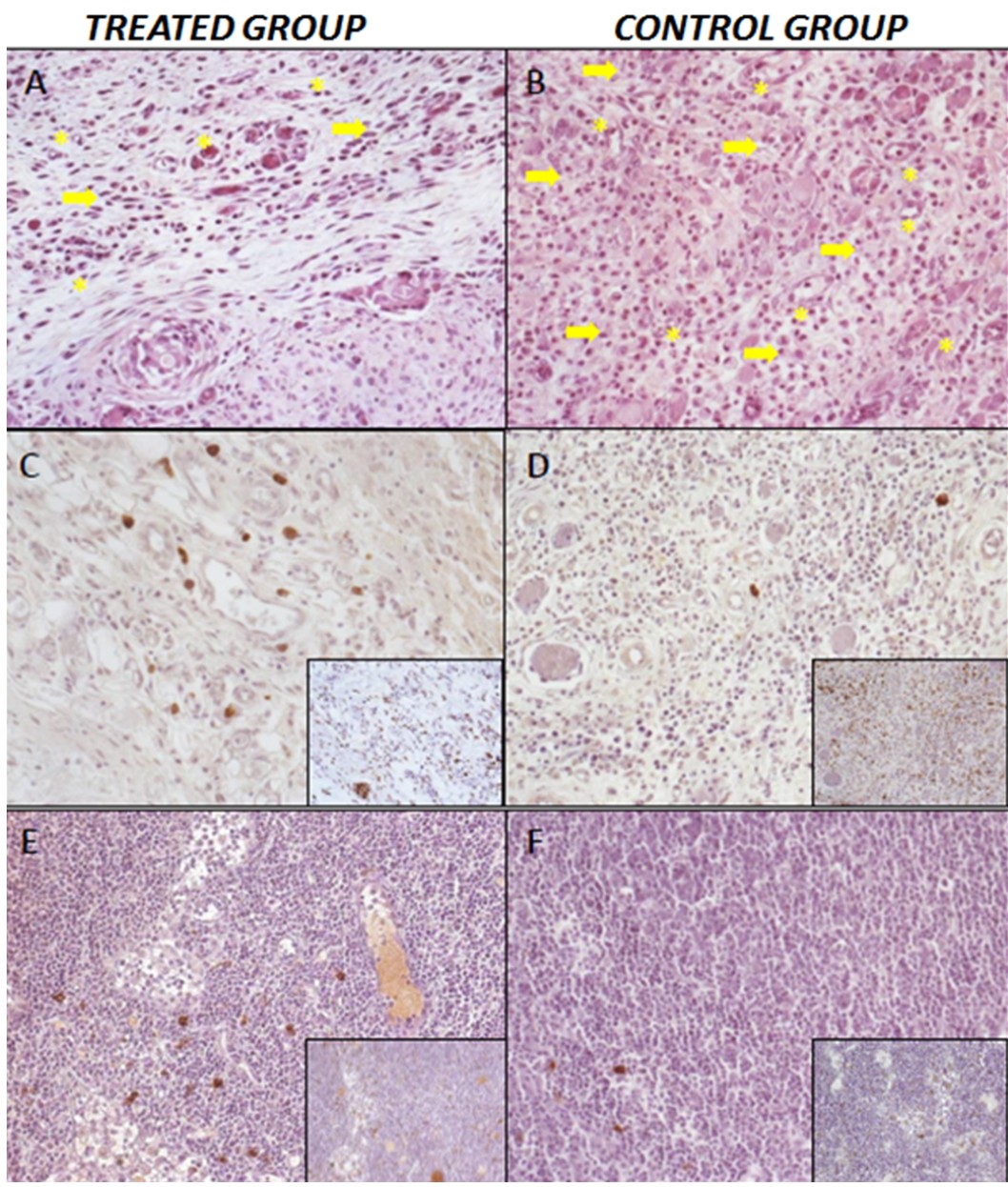

**Figure 5** **Histopathological results.** (A) Treatment group. Low number of inflammatory cells (arrow), capillaries and and neoformed arterioles (asterisk). Hematoxiline-eosine staining, 20×. (B) Control Group l, greater number of inflammatory infiltrate; inflammatory cells (arrow), capillaries and neoformed arterioles (asterisk). Hematoxiline-eosine staining, 20×. (C–D) Expression of macrophagues M1/CD68$^{+}$ in the larger photographs and M2/CD206$^{+}$ expression in the small-right photographs. Cicatricial area of treated group (C) and control group (D), 20×. (E–F) Regional lymph nodes of treated group (E) and control group (F), 20×.

Regarding inflammatory parameters, mast cells recruited to the inflammatory focus are a source of growth factors that stimulate angiogenesis. Their utility in a post-ischemic inflammatory area after acute arterial occlusion has been described elsewhere (*Tajima et al., 2009*). Different chemotactic factors secreted in this focus provoke the migration and activation of mast cells. The density of these in the focus, reaches its highest level at 5 days with a maximum peak at 15 days. These previously described data agree with those obtained in our study.

Per the bibliographic data found, ASCs seem to decrease the inflammatory reaction, as indicated by decreased neutrophil and lymphocyte presence and increased type 2 macrophage infiltration in the early stages (*Ginhoux et al., 2016*).

## CONCLUSIONS

Our results demonstrate that when allogeneic ASCs are applied in the ischemic regions caused by acute arterial occlusion in the extremities of diabetic rats, rejection reactions are not seen. Regulation of the inflammatory parameters is achieved in favor of a better repair of the damaged tissue with excellent tolerance. This gives us the opportunity to make available a treatment option that is safe and seems effective, making it possible to avoid the time involved in the process of isolation, expansion and production of the use of autologous cells.

Our results support the use of allogeneic cell therapy in peripheral arteriopathy; however, we believe that further research will be necessary to clarify its exact mechanisms.

## ACKNOWLEDGEMENTS

The authors want to acknowledgment to the head of Vascular Surgery Department of University Hospital La Paz by the collaboration in the design of the research.

### Funding

This study was funded by Spanish Health Research (a cooperative network for cell therapy research-FEDER (TerCel RD12-0019-0035)). The funders had no role in study design, data collection and analysis, decision to publish, or preparation of the manuscript.

### Grant Disclosures

The following grant information was disclosed by the authors:
Spanish Health Research: TerCel RD12-0019-0035.

### Competing Interests

Prof. D. García Olmo and Dr. M. García Arranz have applied for two patents related to Adipose Derived Mesenchymal Stem Cells titled "Identification and Isolation of Multipotent Cells from Non-Osteochondral Mesenchymal Tissue" (WO 2006/057649) and "Use of Adipose Tissue-Derived Stromal Stem Cells in Treating Fistula" (WO

2006/136244). The remaining authors have no other financial or competing interests to declare.

## Author Contributions

- Jesús Álvarez García performed the experiments, analyzed the data, contributed reagents/materials/analysis tools, wrote the paper, prepared figures and/or tables, reviewed drafts of the paper.
- Soledad García Gómez-Heras conceived and designed the experiments, performed the experiments, analyzed the data, contributed reagents/materials/analysis tools, wrote the paper, prepared figures and/or tables, reviewed drafts of the paper.
- Luis Riera del Moral and Damián García-Olmo conceived and designed the experiments, analyzed the data, reviewed drafts of the paper.
- Carlota Largo performed the experiments, reviewed drafts of the paper.
- Mariano García-Arranz conceived and designed the experiments, performed the experiments, analyzed the data, contributed reagents/materials/analysis tools, wrote the paper, reviewed drafts of the paper.

## Animal Ethics

The following information was supplied relating to ethical approvals (i.e., approving body and any reference numbers):

The protocol was approved by the Animal Welfare Ethics Committee of University Hospital La Paz (No. CEBA/12-14).

## Data Availability

The raw data has been supplied as a Supplemental File.

## Supplemental Information

Supplemental information for this article can be found online at http://dx.doi.org/10.7717/peerj.3664#supplemental-information.

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
