# Peer review of "The effects of allogenic stem cells in a murine model of hind limb diabetic ischemic tissue"

_PeerJ, doi:10.7717/peerj.3664_

## Round 0.1 · original submission · Major Revisions

I think comments raised by both reviewers should be taken into account.

·

Basic reporting

In general the manuscript is well written and it is structured according to the journal style. The manuscript contains the appropriate sections, however some of them need to be worked. Here are some points that may require attention:
1. Abstract. Last sentence-Authors refer to autologous not to allogenic cells.
2. Homogenize the use “allogenic” or “allogeneic” throughout the document.
3. Define ASC, DMSO, FBS, DMEM, PBS, etc. at first mention in the document.
4. Line 60. Properly cite to “P. Zuk”.
5. Line 69. What does mean MMII?
6. Line 84. Change “streptomicym” to “streptomycin”. Provide the concentration of the antibiotics used.
7. Line 111. Is anti-CD31 mono or polyclonal antibody?
8. Line 133. Correct “Macrospcopic”.
Line 150. These values does not match with values in Table 2.
Lines 151-152. Indicate whether the number of macrophage type 2 cells is higher or lower in the treated versus the control group, not only to say that the number is statistically significant.
Lines 158-160. Again, describe the differences observed between the study groups. For example: The number of cell type “X” is higher in group “X”. The number of cell type “Y” is lower in group “Y”.
Lines 120-122. These magnifications are different to those mentioned in Table 2.
Figures are relevant and high quality, however, they need to be better described. Standard deviations are not shown in Figure 3 nor in Table 2. Figure 3 does not show units in X and Y axes. Figures does not contain a description (legend). In Figure 5, to which group corresponds the left and right side?

Experimental design

In Methods include a description of diabetes induction with streptozotocin as well as the method of how ASCs are obtained from adipose tissue. Also, include a subsection describing how macroscopic evaluation was performed and indicating which parameters were measured to assess both ischemic clinical scale and general appearance.
Regarding the immunohistochemical studies (lines 120-122), provide more information about the number of slides produced and analyzed in the study. Was produced one slide per animal, analyzed twenty fields per slide and then averaged the whole group?

Validity of the findings

This manuscript reports the effects of ASCs in a rat model of hind limb diabetic ischemic tissue. It is mentioned that macroscopically, cell-treated rats presented better general and lower ischemic clinical status, and histologically, a better trend towards angiogenesis, greater infiltration of type 2 macrophages and a shortening of the inflammatory process. These findings are certainly very interesting and relevant to peripheral arterial disease treatment with cell therapy. Macroscopic evidences in Figure 4 are impressive. However, I have some doubts about how the clinical evolution was assessed. In Figure 3, what is exactly graphed?
Control group included 12 rats while treated group included 17 rats. However, figures do not seem to represent the whole groups. Therefore, it would be useful to include raw data (primary data) generated during the follow up (clinical evolution and histopathological studies) to better understand the results.

Reviewer 2 ·

Basic reporting

The authors have looked at transplanting adipose tissue derived MSC cell therapy in a diabetic PVD model.

Experimental design

Poorly designed study with no blood flow parameters in a PVD model.

Histology alone with description of the wound area is not sufficient evidence for collateral vessel formation. Please perform a laser Doppler blood flow.

Validity of the findings

Findings need to be presented more systemically

---

## Round 0.2 · accepted · Accept

It is good to see that you have satisfactorily addressed all concerns raised by both reviewers. I am confident you found these very useful.

·

Basic reporting

No comment.

Experimental design

No comment.

Validity of the findings

No comment.

Reviewer 2 ·

Basic reporting

No comment

Experimental design

No comment

Validity of the findings

No comment

Additional comments

No comment